# Rapid geomagnetic changes inferred from Earth observations and numerical simulations

Christopher J. Davies [1] ✉ & Catherine G. Constable [2]

Extreme variations in the direction of Earth's magnetic field contain important information regarding the operation of the geodynamo. Paleomagnetic studies have reported rapid directional changes reaching $1°\,\text{yr}^{-1}$, although the observations are controversial and their relation to physical processes in Earth's core unknown. Here we show excellent agreement between amplitudes and latitude ranges of extreme directional changes in a suite of geodynamo simulations and a recent observational field model spanning the past 100 kyrs. Remarkably, maximum rates of directional change reach ~$10°\,\text{yr}^{-1}$, typically during times of decreasing field strength, almost 100 times faster than current changes. Detailed analysis of the simulations and a simple analogue model indicate that extreme directional changes are associated with movement of reversed flux across the core surface. Our results demonstrate that such rapid variations are compatible with the physics of the dynamo process and suggest that future searches for rapid directional changes should focus on low latitudes.

[1] School of Earth and Environment, University of Leeds, Leeds LS2 9JT, UK. [2] Institute of Geophysics and Planetary Physics, Scripps Institution of Oceanography, University of California at San Diego, La Jolla, CA 92093-0225, USA. ✉email: c.davies@leeds.ac.uk

The large-scale secular variation of Earth's internally generated magnetic field is now reasonably well-established by global models based on observations spanning the past two decades[1] and the historical period[2]. Prominent features of these models, such as the difference in activity between Atlantic and Pacific hemispheres and rapid changes at high latitudes, have been linked, respectively, to thermal interactions between the core and mantle[3,4] and accelerating jets in the core[5], and hence to the operation of the geodynamo. Over longer timescales, paleomagnetic studies[6–8] have identified changes in field intensity $B$ in the Levantine region around 1000 BCE of $\partial B/\partial t \approx 0.75$–$1.5\,\mu T\,yr^{-1}$ that are significantly faster than the largest values of $\partial B/\partial t \approx 0.12$ $\mu T\,yr^{-1}$ for the modern field[9] and averages over the Holocene field[10]. Although the nature and origin of this 'intensity spike' has proved controversial[11–13], recent estimates of its rate of change[8,12] are compatible with bounds based on the kinematics of core flow[14], and apparent inconsistencies in the available data around 1000 BCE can be explained by large age uncertainties for some samples[15]. Additional support for intensity spikes has come from studies in China[16] and Texas[17] and from numerical simulations[13] that reproduce similar values of $\partial B/\partial t$. In the simulations, intensity spikes reflect the migration of normal-polarity flux patches across the core surface[13].

Rapid changes in field direction have also attracted significant attention, particularly in the context of polarity reversals[18]. Historically, the fastest directional changes were attributed to lava flows at Steens Mountain, though these results are now thought to be untenable[19]. Currently, the fastest directional changes noted are those recorded by sediments in central Italy[20], where angular changes in the Virtual Geomagnetic Pole position ($\hat{\mathbf{P}}_V$) reach values of $\partial \hat{\mathbf{P}}_V/\partial t \sim 1°\,yr^{-1}$. These rates are about a factor of 10 faster than values of $\partial \hat{\mathbf{P}}_V/\partial t \sim 0.1°\,yr^{-1}$ for the modern field, similar to the difference in rates of intensity change between the modern field and spikes. The reliability of the paleomagnetic analysis of the Italian sediments is presently the subject of active debate[21,22]. Overall, it has proved difficult to substantiate extremely rapid changes in individual paleomagnetic records either during reversals or at other times. Furthermore, the compatibility of these directional changes with the physics of the dynamo process, and the dynamical origin of such variations, has not been investigated.

Rapid directional changes have also been obtained in simple kinematic field models. Brown et al.[23] independently reduced the axial dipole field coefficient in the CALS7k.2 Holocene field model[24] to simulate synthetic reversals. They found that reversal duration varied significantly with location, with certain localities such as Tahiti experiencing rapid changes in the polarity of the VGP. Fournier et al.[11] constructed a synthetic model of an intensity spike at mid-latitudes and noted that movement of the spike can change the dipole latitude by ~1°\,yr^{-1}. However, it is still not known whether these rates represent a "speed limit" for the field, whether such changes exhibit systematic geographical preference, or how they relate to core dynamics.

In this study, we seek to establish how fast the local direction of the geomagnetic field might change in general and whether rapid changes occur at preferred locations on Earth's surface. Furthermore, we investigate the processes at the core surface that give rise to the most rapid directional changes. To do this, we take advantage of recent developments in paleomagnetic field modelling and numerical dynamo simulations to document rates of change during stable polarity times as well as excursions and reversals. First, we use GGF100k[25], the first time-varying paleomagnetic field model for the time interval 0–100 ka. Accessing such long timescales is important because the fastest field changes are by definition rare events. Our results are bolstered by higher

resolution short term models spanning the Laschamp excursion. Second, we extend a set of tools that were developed to study rapid intensity changes in dynamo simulations[13]. This allows us to compare the whole spectrum of directional changes between simulations and observational field models.

## Results

**Models and simulations of rapid directional changes**. We search for extreme changes in field direction in GGF100k and in a suite of 16 dynamo simulations that produce a range of dynamical behaviour and reproduce various features of the geomagnetic field (see ref. [13] and Methods). Following previous work[26,27], we compare simulations in terms of the magnetic Reynolds number $Rm$, which is used as the unique identifier in both tables and figures. Our simulations span the range $Rm = 100$–$700$ (compared to the value for Earth's core of $Rm \sim 10^3$), and some exhibit polarity reversals. To ensure robust results for comparing with the paleomagnetic analyses, we consider two field properties, the local field vector $\mathbf{B}$ and its transformation to the equivalent Virtual Dipole vector $\mathbf{P}_V$ with unit vectors denoted by $\hat{\mathbf{B}}$ and $\hat{\mathbf{P}}_V$, respectively. For both quantities, the corresponding rate of change (labelled $\partial \hat{\mathbf{P}}_V/\partial t$ and $\partial \hat{\mathbf{B}}/\partial t$) is calculated using the difference between values at time $t$ and $t + \Delta t$. We find the maximum rate of change of directions at each location on a 2° by 2° latitude–longitude $(\lambda, \phi)$ geographic grid (Methods). The largest changes anywhere on the globe are referred to as 'extremal events' and are denoted with a subscript ex.

Figure 1 shows the geographic variability in $\partial \hat{\mathbf{P}}_V/\partial t$ and time-series at the locations of $(\partial \hat{\mathbf{P}}_V/\partial t)_{ex}$ for one simulation ($Rm = 386$) that undergoes excursions but not reversals, one reversing simulation ($Rm = 450$), and the 0–100 ka time-varying field model GGF100k[25]. (Time-series for all 16 simulations studied are provided in Supplementary Fig. 1). The simulations shown in Fig. 1 have been run for many dipole decay times (see Supplementary Table 1 for details) and have complete spatial and temporal coverage, thus providing a statistically representative set of extreme rates of change that occur at different times and places across the globe. The dominant event for GGF100k is around the Laschamp excursion at 41 ka. The regional focus and relatively smooth time variation in GGF100k reflect both the shorter time span, and the uneven temporal and spatial coverage of the underlying paleomagnetic data. Note that the temporal sampling, $\Delta t$, and size of the spatial grid are chosen to ensure that our estimates of extreme rates of change are conservative. Experimentation with denser spatial and temporal sampling and with LSMOD.2, a shorter higher resolution model[28] spanning the Laschamp excursion (Supplementary Fig. 8), suggests the possibility of extreme values a factor of 2–5 larger than those shown for GGF100k. Nevertheless, both reversing and non-reversing simulations as well as GGF100k exhibit rapid directional changes with $\partial \hat{\mathbf{P}}_V/\partial t > 1°\,yr^{-1}$ (see also Fig. 2).

Surprisingly, most of our simulations produce extremal directional changes that are removed by more than 5 kyr from reversals (see, e.g. Fig. 1, middle panel, where for $Rm = 450$, the extremal change precedes the equatorial crossing of the axial dipole by at least 5 kyr). Similar behaviour occurs in GGF100k, where the extremal directional change occurs slightly after the main phase of the Laschamp excursion. Another common feature of extreme directional changes is a considerably diminished overall field strength, but not necessarily a local minimum in $B(t)$ (Fig. 1). It is notable that while all reversing simulations produce high rates of directional change $>1°\,yr^{-1}$, even non-reversing simulations can still produce similarly high values associated with global or regional excursions and intensity minima.

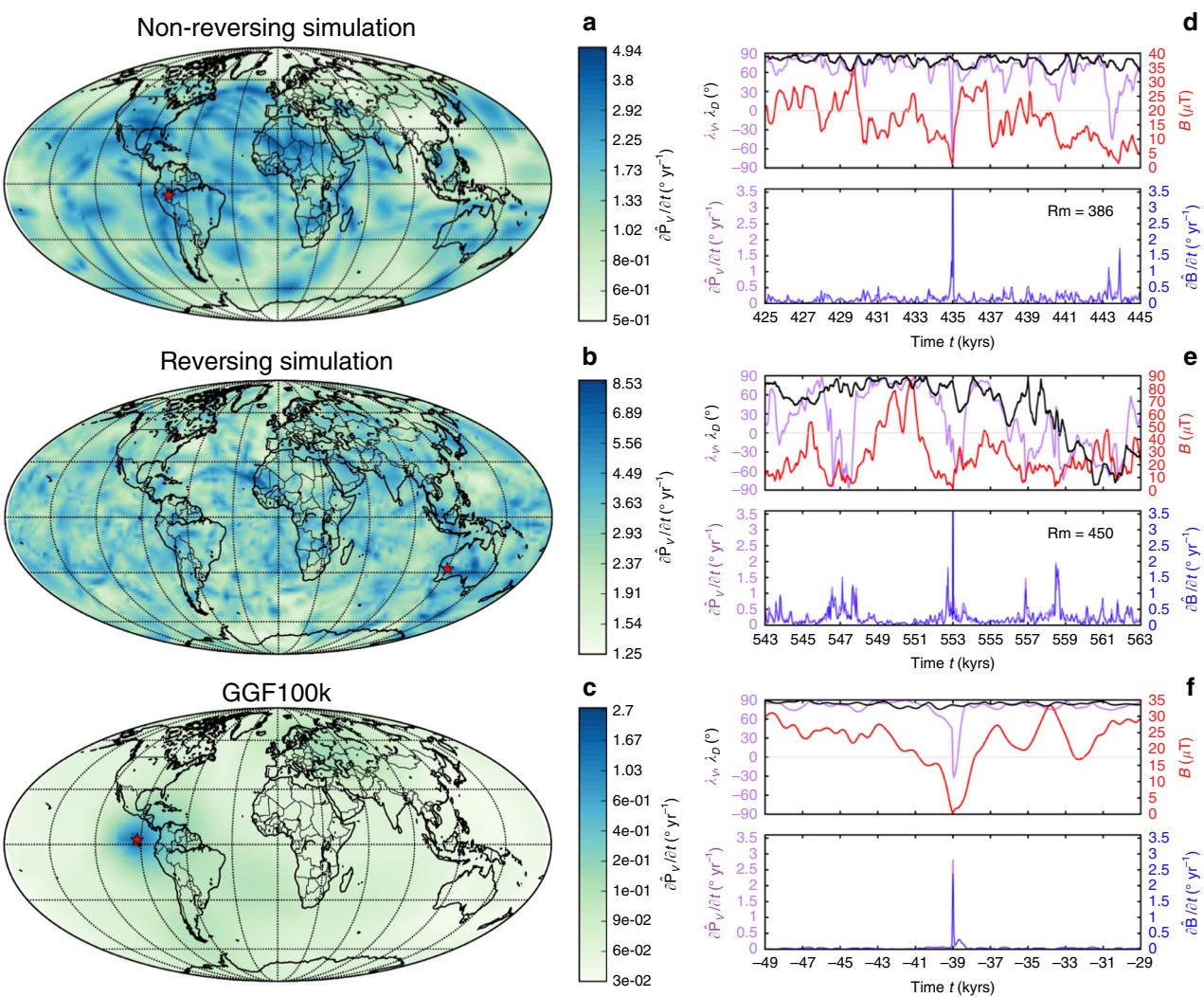

**Fig. 1 Rapid directional changes in two geodynamo simulations and GGF100k.** Shown are a non-reversing simulation with magnetic Reynolds number $Rm = 386$ (**a**, **d**), a reversing simulation with $Rm = 450$ (**b**, **e**) and the observational field model GGF100k[25] (**c**, **f**). Left column shows Mollweide projections at Earth's surface of the largest change in VGP position, $\partial\hat{\mathbf{P}}_V/\partial t$, as a function of location in ° yr$^{-1}$. Red stars show the location of $(\partial\hat{\mathbf{P}}_V/\partial t)_{ex}$ on each plot. Note the different colour scales and that values at each location may not have occurred at the same point in time. Right-hand panels show directional data at the locations of $(\partial\hat{\mathbf{P}}_V/\partial t)_{ex}$ over a 20 kyr period with the extreme event at the midpoint. Here the top row of each panel shows the latitude $\lambda_V$ of $\hat{\mathbf{P}}_V$ (purple), the dipole latitude $\lambda_D$ (black), and the field strength $B$ (red); the bottom row shows $\partial\hat{\mathbf{P}}_V/\partial t$ (purple) and the rate of change of the field vector $\partial\hat{\mathbf{B}}/\partial t$ (blue). Simulations have been run for 232 kyrs for $Rm = 386$ and 415 yrs for $Rm = 450$.

The time-series of $(\partial\hat{\mathbf{P}}_V/\partial t)_{ex}$ and $(\partial\hat{\mathbf{B}}/\partial t)_{ex}$ (Fig. 1, Supplementary Fig. 1) show a clear spike at the times of extreme directional changes. However, the shapes of the corresponding $\hat{\mathbf{P}}_V$ and $\hat{\mathbf{B}}$ time-series vary across simulations: some are spike-like, others have steep increases and decreases with a relatively flat intermediate period, and occasionally the time-series does not show both a rise and fall. We think these distinctive spike-like patterns in the time-derivatives are best described as local or regional rapid accelerations in magnetic field changes. This view is supported by the fact that large values of $(\partial\hat{\mathbf{P}}_V/\partial t)_{ex}$ and $(\partial\hat{\mathbf{B}}/\partial t)_{ex}$ are not necessarily associated with low values of $\lambda_D$, the latitude of the geomagnetic dipole axis, as commonly seen in reversals or field excursions, although in both GGF100k and the various simulations they do generally seem to occur at times when field strength is low enough that non-dipole field activity may dominate. We will just call them extremal directional changes.

Figure 2a shows the maximum rates of change in field direction for all simulations. Remarkably, extremal directional changes can

reach 10° yr$^{-1}$ in the simulations. This is ten times faster than the fastest variations so far reported in individual paleomagnetic records[20] and about 40 times faster than the fastest changes in Holocene field models CALS10k.2 and HFM.OL1.A1[10]. However, the examples in Fig. 1 are quite compatible with our conservative estimates of peak rates of change of 2.5–3.5° yr$^{-1}$ found in GGF100k and LSMOD.2 in the millennia surrounding the Laschamp excursion when the dipole moment is very low.

The absolute latitude corresponding to the most extreme directional changes, $|\lambda_{ex}|$, is generally <40° in all simulations (Fig. 2b). Using either $\hat{\mathbf{B}}$ or $\hat{\mathbf{P}}_V$ to measure directional changes gives broadly similar results, though stronger latitudinal variability arises when considering $\hat{\mathbf{B}}$ compared to $\hat{\mathbf{P}}_V$, with noticeably faster directional changes tending to occur at lower latitudes. Figure 3a shows the histogram of $\partial\hat{\mathbf{B}}/\partial t$ values for GGF100k, which is well approximated by a log-normal distribution both globally and at individual latitudes. The corresponding fits to cumulative distribution functions (CDFs) at different latitudes for GGF100k (Fig. 3b) clearly show that the probability of observing

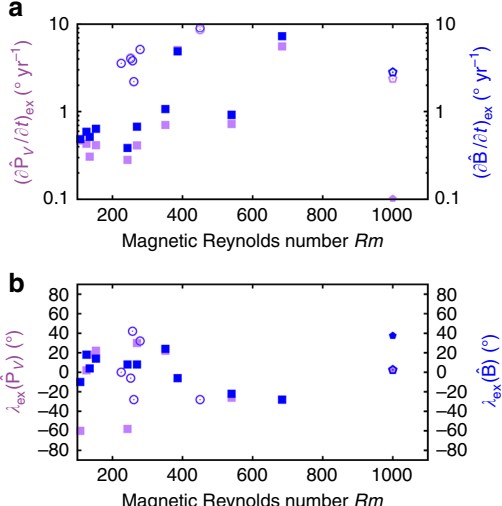

**Fig. 2 Amplitudes and locations of extreme directional changes in all simulations and GGF100k.** The field vector is denoted $\hat{\mathbf{B}}$ and the VGP position is denoted $\hat{\mathbf{P}}_V$. In **b** the locations of maximum rates of change of $\hat{\mathbf{B}}$ and $\hat{\mathbf{P}}_V$ are $\lambda_{ex}(\hat{\mathbf{B}})$ and $\lambda_{ex}(\hat{\mathbf{P}}_V)$, respectively. Closed (open) symbols indicate non-reversing (reversing) simulations. Where shown, values for $Rm = 1000$ are averages from the Holocene field model CALS10k.2[10] (solid symbols) and GGF100k[25] (open symbols), otherwise the values plot below the lower limit on the ordinate. All locations are given in degrees.

faster rates of directional change is greatest at the lower latitudes. A synthesis of the scale and shape information presented in Fig. 3b is shown in Fig. 3c, d, which show ratios of the mean ($\mu$) and standard deviation ($\sigma$) of the log-normal distributions at 40° and 80° latitude to the values at 0° for GGF100k and all of the dynamo simulations. Normalised mean values decrease significantly with increasing latitude, indicating that the CDF curves at lower latitudes are shifted towards higher values of $\partial\hat{\mathbf{B}}/\partial t$, thus following the behaviour of the GGF100k model (Fig. 3b). Standard deviations (reflecting relative variability in $\partial\hat{\mathbf{B}}/\partial t$) decrease with latitude in most simulations, as in GGF100k, whereas a few simulations with $Rm \approx 200$–$300$ show maximum values of $\sigma$ at mid-to-low latitudes. Overall, the excellent agreement between GGF100k and the dynamo simulations shows a general preference for rapid directional changes at lower latitudes. These variations in $\hat{\mathbf{B}}$ are readily linked to core processes as will be seen in the next section, where we outline the strategy for investigating the origin of rapid directional changes.

**Linking rapid directional changes and core processes.** We first consider a simple model of rapid directional changes in which the radial CMB field consists of an isolated moving flux patch superimposed on a static axial dipole field defined by the Gauss coefficient $g_1^0 = 80\,\mu\text{T}$. The spatial form of the patch is described by a Fisher–von Mises probability distribution function with amplitude $A$ and half width $\sigma_M = 15°$ (Methods). The patch is moved in latitude (longitude) using constant intervals of $\delta\lambda$ ($\delta\phi$).

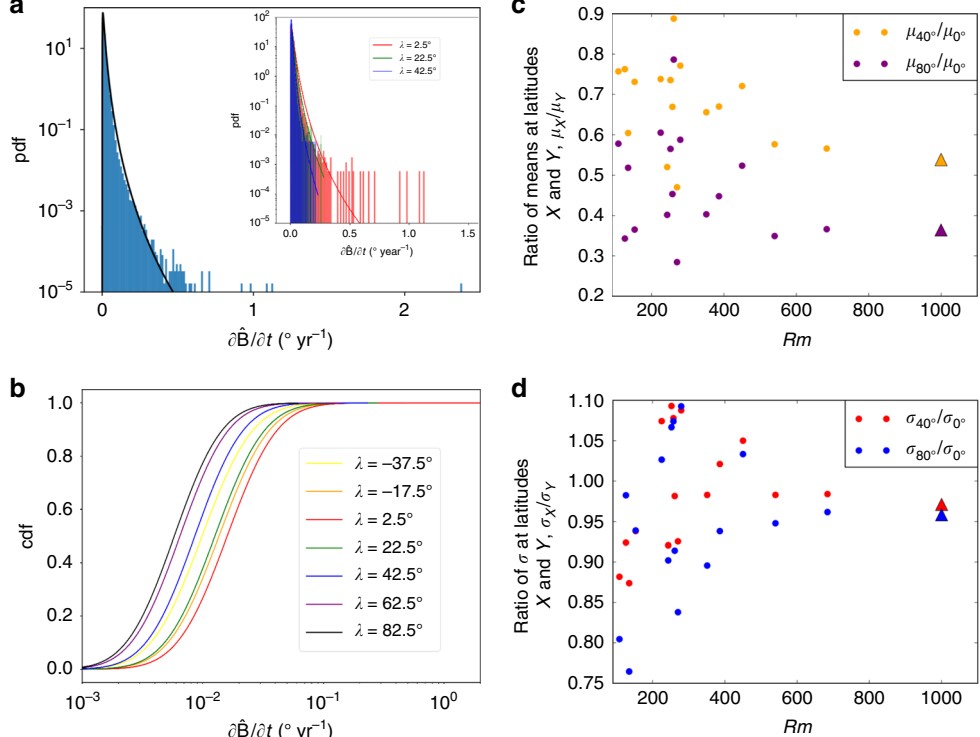

**Fig. 3 Statistics of extreme directional magnetic field changes. a** Normalised histogram for rates of change of the field vector $\hat{\mathbf{B}}$ in the GGF100k model[25] sampled on a 2° × 2° latitude–longitude grid at 100 year time intervals. The black line shows the fit of a log-normal probability density function (PDF) to the data with mean $\mu = 0.009$ and standard deviation $\sigma = 0.011$. The inset shows histograms at latitudes of 2.5° (red), 22.5° (green) and 42.5° (blue) with PDFs shown as solid lines. **b** Cumulative distribution functions (CDFs) of $\partial\hat{\mathbf{B}}/\partial t$ at different latitudes (shown by colours) from the GGF100k model, showing higher rates of change at lower latitudes. **c** The ratio of the mean values of PDFs at two different latitudes (denoted by subscripts) for GGF100k and all dynamo simulations as a function of the magnetic Reynolds number Rm. GGF100k is shown at $Rm = 1000$ as in Fig. 2. Panel **d** is the same as **c** but showing ratios of standard deviations.

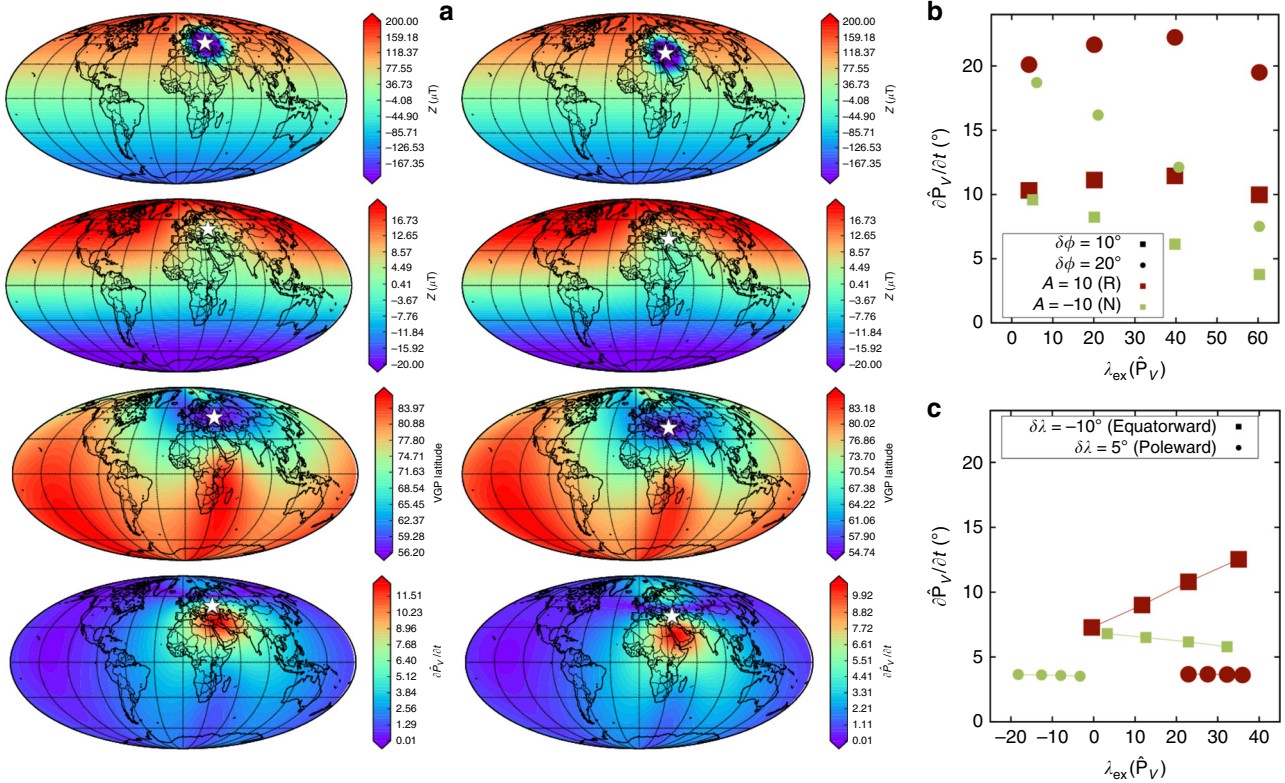

**Fig. 4 Directional changes produced by a simple model of a patch of radial field moving across the core surface. a** The effect of a reversed flux patch of amplitude $A = 10\,\mu$T with an initial latitude of 60° moving southward to 50° (left) and then to 40° (right) ($\delta\lambda = -10°$). Rows show, respectively, the vertical magnetic field at the CMB and surface, $Z$, VGP latitude at the surface, and the rate of change of VGP position $\partial\hat{\mathbf{P}}_V/\partial t$. The centre of the patch on the CMB is denoted by the star. **b, c** The amplitudes, $\partial\hat{\mathbf{P}}_V/\partial t$, and locations, $\lambda_{ex}(\hat{\mathbf{P}}_V)$, of the fastest directional changes for normal ($A < 0$, denoted N) and reversed ($A > 0$, denoted R) polarity patches. In **b**, patches initially at 5°, 20°, 40° and 60° latitude are moved longitudinally in increments of $\delta\phi = 10°$ (squares) and 20° (circles). In **c** patches initially placed at 40° longitude are moved equatorward and poleward in increments of $\delta\lambda = -10°$ and $\delta\lambda = 5°$, respectively. In all cases in **b, c** $\partial\hat{\mathbf{P}}_V/\partial t$ is largest for reversed patches.

Time increments are arbitrarily set to $\Delta t = 1$ yr and so this model cannot provide an absolute rate of directional change; nevertheless it allows us to systematically compare the influence of isolated normal ($A < 0$) and reversed ($A > 0$) CMB patches on directional changes at the surface without the complication of multiple patches as arises in the simulations.

Figure 4 shows an example evolution of the CMB and surface fields and a synthesis of models with normal and reversed patches moving in various directions. The main conclusion is that reversed patches produce faster directional changes than normal patches, all other factors being equal (compare red, reverse, and green, normal, points in Fig. 4b, c). Interestingly, normal-polarity patches moving towards the pole can induce a change in $\partial\hat{\mathbf{P}}_V/\partial t$ in the opposite hemisphere that is almost as large as the change induced by the reversed patches in the same hemisphere (Fig. 4c, circles). However, in a more realistic scenario where the flux patch executes some combination of the motions shown in Fig. 4 the fastest changes, which are the main focus here, would clearly arise from reversed flux migration. In this model the extremal events arise in regions where the amplitude of the reversed flux balances that of the dipole field at the surface (Fig. 4a), corresponding to a region of low inclination. The non-dipole field, which is weaker and more variable than the dipole field, then controls the rate of directional change.

This simple model suggests that reversed flux patches could be relevant for understanding rapid directional changes in the simulations. Figure 5 shows snapshots of $\lambda_V$, the latitude of $\hat{\mathbf{P}}_V$, together with the vertical magnetic field at the core-mantle boundary (CMB) and at Earth's surface for a representative simulation (see Supplementary Information for other cases). The extremal event arises as a reversed flux patch moves southwards beneath the northeast Asian region. This patch, which is clearly visible on the CMB and also apparent in the surface field, creates a local inclination anomaly of opposite sign compared to the surrounding field, which is reflected in $\lambda_V$.

To investigate this behaviour in more detail, we have developed a tool for locally reducing the field strength in regions surrounding extreme events (Methods). We consider four $30° \times 30°$ quadrants with a common vertex at the location of the extremal event and halve the CMB field strength in each quadrant in turn. We then compute directional changes at the surface based on these CMB fields as before. Results from three simulations are shown in Fig. 6 (see Supplementary Figs. 2–6 for more examples). Figure 6a shows the result of performing this analysis on the simulation in Fig. 5, which reveals that the northeast quadrant containing the large reversed flux patch makes the biggest contribution to the observed rate of change. In Fig. 6b, reversed flux patches in the northwest and southwest quadrants that lie either side of the equator are the cause of the rapid change. In Fig. 6c, the largest contributions to the directional change arise from a reversed flux patch moving towards the observation point from the northeast.

This analysis is somewhat simplified because it does not account for the changing direction and strength of different patches. Possible extensions to our method include tracking individual patches[29] and using time-dependent factors for

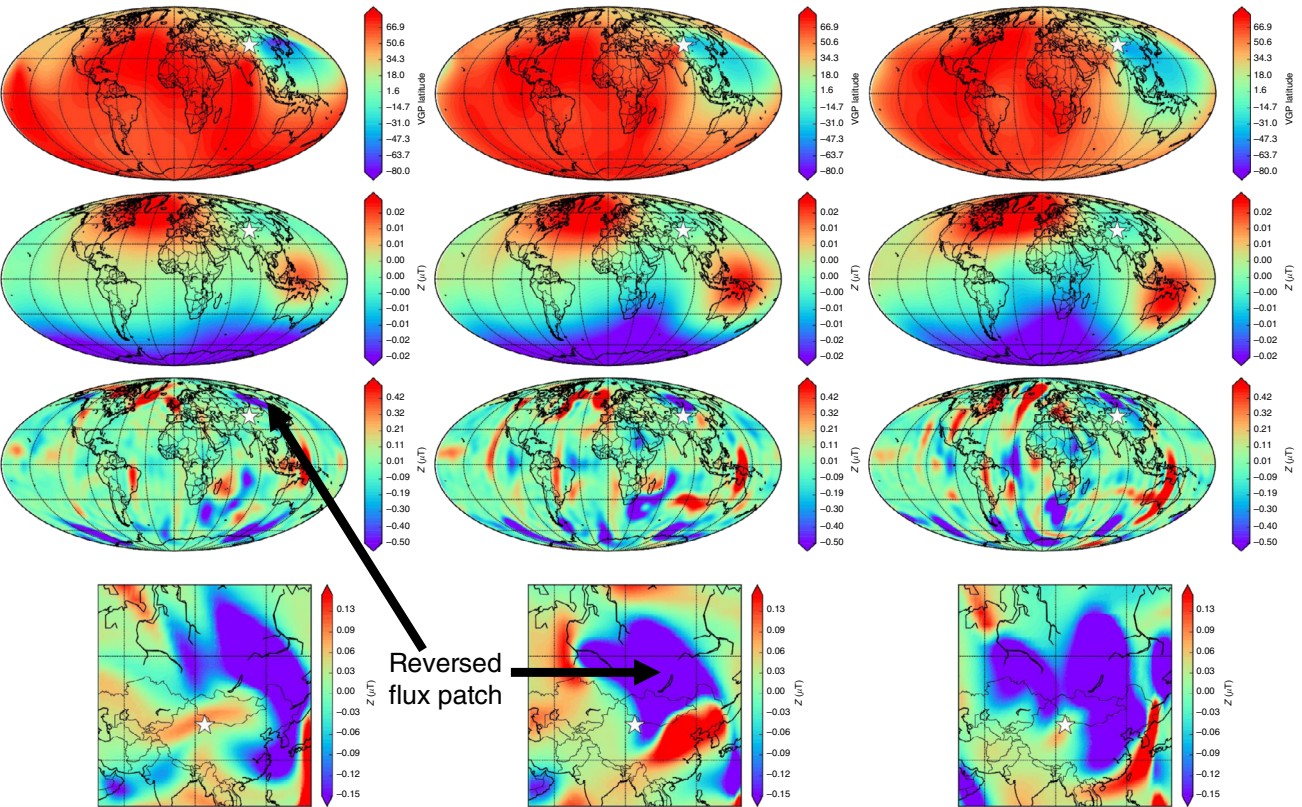

**Fig. 5 Evolution of an extreme change in the VGP position, $\hat{P}_V$, for the simulation with $E = 5 \times 10^{-4}$, Pm $= 5$, Ra $= 400$ and Rm $= 257$.** Rows show the VGP latitude $\lambda_V$, vertical component of the magnetic field $Z$ at Earth's surface, and at the CMB, and a local Mercator projection of the CMB field around the location of maximum change (white star). Columns show times just before (left), during (centre) and just after (right) the extreme change. The extremal event arises as a reversed flux patch (blue) migrates from the northeast towards the point of maximum change.

reducing the local field strength as the simulation evolves, though both options have their drawbacks. Nevertheless, we believe that the results in Fig. 6, combined with the results from the simple patch model (Fig. 4), present a strong link between rapid directional changes at Earth's surface and the movement of reversed flux patches across the core surface.

## Discussion

Our suite of geodynamo simulations span a wide range of physical parameters and access reversing and non-reversing dynamo regimes; however, as with all current models, they still do not reach the low values of the Ekman number $E$ and magnetic Prandtl number Pm that characterise Earth's core. However, all exhibit some Earth-like features, and we find consistent results for the locations and amplitudes of extremal events and no obvious dependence on Rm. As discussed in detail in our previous work on intensity spikes[13], there is no reason to believe that the rates of change in these simulations should be greater or smaller than those that would be obtained at more extreme parameters. Indeed, we find that different values of $E$ and Pm produce similar rates of change in the same dynamo regime (e.g. stable dipolar or reversing).

The GGF100k model is limited by the spatial and temporal distribution of the data and the accuracy of the chronological constraints at different locations. This produces a smoothed record in time that can only resolve large-scale spatial features. We expect these limitations to result in estimates of field changes that are lower than the actual variations. This is confirmed by the fact that higher rates of change are evident in LSMOD.1 and LSMOD.2, two recent higher resolution global time-dependent

field models spanning 50–30 ka[28,30] that incorporate updated chronological information. The use of the longer record provided by GGF100k reveals that the most extreme values occur at times of low field strength and the approximately log-normal distributions of rates of change further demonstrate that extreme values are relatively rare over the past 100 kyr. From these considerations and the excellent consistency between the locations and amplitudes of extreme directional changes in both observational and numerical models, which are entirely independent, we believe our results provide strong evidence that the geodynamo can produce much faster directional changes than have previously been considered viable.

The fastest directional changes are of comparable magnitude in both simulations and paleofield models, reaching ~10° yr$^{-1}$ in simulations and rising to 4.8° and 22.5° yr$^{-1}$ in GGF100k and LSMOD.2, respectively, when these models are more densely sampled at an interval of 10 years. This is faster than the latest published local paleomagnetic estimates[20] of 1° yr$^{-1}$, and suggests that such rapid directional changes must be fully compatible with the physics of the dynamo process. Of course, such events are exceedingly rare: in the 100 kyr spanned by GGF100k, we found only one time interval with directional changes of >2° yr$^{-1}$ from over 5 million samples of the field, whereas 95% of the samples have rates of change below 0.05° yr$^{-1}$ (Fig. 3). However, faster rates of directional change tend to occur at lower latitudes where the field is weaker. Our results suggest that the most rapid directional changes occur at latitudes <40°, which could be helpful in guiding future paleomagnetic acquisitions.

The best evidence for rapid directional changes in time-varying paleofield models comes from times that are close to known excursions. Although rapid changes are not necessarily associated

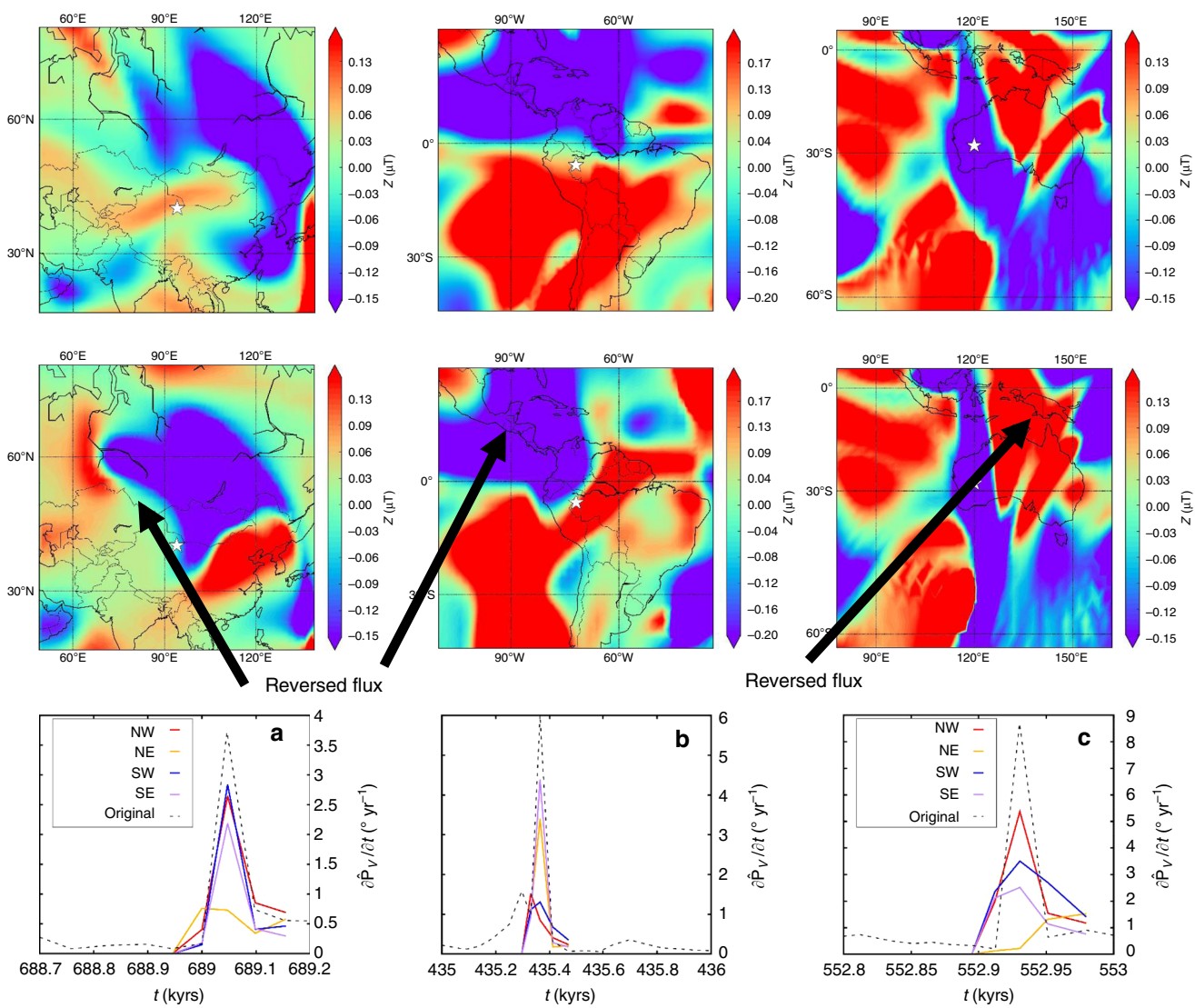

**Fig. 6 Contributions of CMB field features to rates of directional change at Earth's surface.** The three columns show extreme changes arising in the northern hemisphere (left), equatorial region (centre) and southern hemisphere (right) in simulations with $E = 5 \times 10^{-4}$, Ra = 400, Pm = 5, $Rm = 257$ (**a**); $E = 5 \times 10^{-4}$, Ra = 250, Pm = 10, $Rm = 386$ (**b**); $E = 5 \times 10^{-4}$, Ra = 350, Pm = 10, $Rm = 450$ (**c**). Rows show local Mercator projections of the radial CMB field just before (top) and during (middle) an extreme directional change located at the star. The bottom row shows the rate of VGP change, $\partial\hat{\mathbf{P}}_V/\partial t$, at the location of the star based on the original CMB field and four modified versions of the CMB field wherein the field in the denoted quadrant has been halved in amplitude. In this projection the four quadrants denoted northwest (NW), northeast (NE), southwest (SW) and southeast (SE) tessellate the regions shown in the first two rows and share a common vertex centred at the star.

with large deviations of the dipole axis, they are often associated with a decrease in field strength that accompanies excursions[25,30]. However, we do not yet rule out the possibility that large directional changes might occur during stable polarity times due to rapid growth of a reverse flux patch that might not lead to global large-scale decrease in intensity. Such a scenario is seen in a number of our dynamo simulations (e.g. Supplementary Fig. 1e, g, h, k) and may correspond to a so-called regional rather than global excursion. The quality of the current paleomagnetic record is not adequate to map such regional behaviour in detail, but recent progress in global field modelling suggests it can be achieved in the future.

In the simulations, extremal directional changes arise from the migration of reversed flux patches towards the observation site, which produces a sharp change in direction. Reversed flux patches tend to occur more frequently in the equatorial region, which explains the observation that rapid directional changes occur at low latitudes. In our simple model of an isolated CMB

flux patch the fastest directional changes are always associated with migration of reversed flux on the CMB; however, normal flux patches seem likely to have a general role in producing slower variations. Examining the relative contributions of normal and reversed CMB flux across the statistical distribution of directional changes is a significant undertaking owing to the detailed analysis required for each event, but could be approached based on the tools developed here to study extremal events.

We can compare the characteristics of directional changes to those of intensity variations, which were previously analysed[13] in 9 of the 16 simulations presented here. Rapid and localised changes in intensity comparable to the rates inferred for the Levantine spike were found to occur predominantly at latitudes >50° and were produced by rapid migration and intensification of normal-polarity flux patches on the core surface. These results suggest that extremal intensity and directional variations reflect different physical processes at the top of the core. It is of course well-known that the local surface inclination and declination

sample a different part of the CMB from the intensity. For an axial dipole field inclination sampling at the CMB is biased toward lower latitudes (except at the equator), declination preferentially samples east and west of the sampling location, and local surface intensity sampling peaks poleward of the sampling site[31]. Thus spatial sampling bias may further reinforce the ability to detect latitudinal differences between extremal events in direction and intensity. Further work is needed to assess whether under some circumstances accounting for this sampling bias may allow any information about intensity spikes to be extracted from local surface directional changes.

New global time-dependent field models for periods preceding the Holocene[25,28,30] are revealing geomagnetic field behaviour that is not represented in the relatively short historical record. The strong agreement between rates and locations of rapid directional change in these models and numerical simulations means that it is now possible to probe the dynamical origin of these extreme events. It is plausible that the characteristics of extreme events contain information about the structure and dynamics of the CMB region. A stably stratified layer in the uppermost core, which has been inferred from seismology[32] and may reflect thermal[33] or chemical[34,35] interactions with the mantle, may suppress radial motions and weaken the field strength at the core surface[36]. On the one hand a weaker field is generally associated with increased rates of change in our simulations; however, we also find that extremal directional changes arise from horizontal motion of flux patches, which is not hindered by stratification. Lateral variations in CMB heat flow can induce longitudinal variations in the structure of the magnetic field[37] and secular variation[38] and may even produce regional stratification[39]. Although we observe no clear differences between the 5 simulations that include CMB heat flow variations and the 11 that do not, this sample is not large enough to allow robust conclusions on this point. Future work linking numerical simulations to global models of the paleomagnetic field can help resolve these issues.

## Methods

**Geodynamo simulations**. Full details of the numerical simulations are given in our previous papers[13,27] and so only a brief description is provided here. Our computer code solves the equations that describe conservation of mass, momentum, energy and magnetic flux in a rotating spherical shell. In the dimensionless governing equations the input parameters are the Ekman number $E$, measuring the ratio of viscous to Coriolis effects; the magnetic Prandtl number Pm, the ratio of viscous and magnetic diffusivities; the Rayleigh number Ra, measuring the vigour of convection; the Prandtl number Pr = 1, the ratio of viscous and thermal diffusivities; the ratio of inner to outer boundary radii, $r_i/r_o = 0.35$; and the amplitude of boundary heat flow heterogeneity $q^\star$ (=0 for homogeneous boundaries). At $r_o$ the boundary conditions are no-slip, electrically insulating and fixed heat flux (FF), whereas at $r_i$ the boundary conditions are no-slip, electrically conducting or insulating, and either fixed temperature (FT) or fixed flux. Simulations with a laterally varying heat flux at $r_o$ use the pattern from ref. [40]. The magnetic Reynolds number Rm, the ratio of advection and diffusion of magnetic field, serves to uniquely identify the simulations.

We consider a total of 16 simulations, which are listed in Supplementary Table 1. The simulations with $E = 5 \times 10^{-4}$ and $1.2 \times 10^{-4}$ were originally published in ref. [27], the simulations with $E = 10^{-5}$ were published in ref. [41] and the simulations with $E = 10^{-3}$ were added in ref. [42]. These simulations were chosen because they have been shown to reproduce various features of Earth's magnetic field; however, they have never been used to study local directional changes.

**Analysis of extreme directional changes**. The geodynamo simulations use dimensionless variables (denoted by stars below), which must be converted to physical units in order to be compared with geomagnetic observations. Following our previous work[13], time $t$ is scaled using the advection time, $t = (\text{Rm}_m/\text{Rm}_E) t^\star d^2/\eta$, where $d = 2264$ km is the shell thickness, $\eta = 1 \text{ m}^2 \text{ s}^{-1}$ is the magnetic diffusivity and $\text{Rm}_m$ and $\text{Rm}_E$ are the magnetic Reynolds number of the model and Earth, respectively. Intensity is scaled based on the polar value of the time-averaged field strength $B$, which gives similar results to other choices[13].

At each time the three components $X$, $Y$ and $Z$ of the local magnetic field vector $\mathbf{B}$ are calculated on a 2° by 2° latitude–longitude grid at Earth's surface as described in ref. [13]. From these components, the strength of the local field vector $B = \sqrt{(X^2 + Y^2 + Z^2)}$ and the inclination $I$ and declination $D$ are obtained.

Defining the instantaneous units vectors

$$\hat{X}(t) = X(t)/B(t), \hat{Y}(t) = Y(t)/B(t), \hat{Z}(t) = Z(t)/B(t), \quad (1)$$

the rate of change of $\hat{\mathbf{B}}$ between times $t$ and $t_1 = t + \Delta t$ is given by

$$\frac{\partial \hat{\mathbf{B}}}{\partial t} = \frac{\arccos[\hat{X}(t)\hat{X}(t_1) + \hat{Y}(t)\hat{Y}(t_1) + \hat{Z}(t)\hat{Z}(t_1)]}{\Delta t} \quad (2)$$

and the rate of change of $B$ is

$$\frac{\partial B}{\partial t} = \frac{B(t_1) - B(t)}{\Delta t}. \quad (3)$$

Variations of the virtual dipole vector $\mathbf{P}_V$ are calculated in the same manner as variations in $\mathbf{B}$. The virtual dipole position $\hat{\mathbf{P}}_V$ with latitude $\lambda_V$ and longitude, $\phi_V$ is calculated from $I$ and $D$ and converted to unit vectors according to

$$\hat{X}_V = \cos(\lambda_V)\cos(\phi_V), \hat{Y}_V = \cos(\lambda_V)\sin(\phi_V), \hat{Z}_V = \sin(\lambda_V). \quad (4)$$

The rate of change of $\hat{\mathbf{P}}_V$ between times $t$ and $t_1$ is then given by

$$\frac{\partial \hat{\mathbf{P}}_V}{\partial t} = \frac{\arccos[\hat{X}_V(t)\hat{X}_V(t_1) + \hat{Y}_V(t)\hat{Y}_V(t_1) + \hat{Z}_V(t)\hat{Z}_V(t_1)]}{\Delta t}. \quad (5)$$

The amplitude of the virtual dipole vector $P_V$ is the virtual dipole moment, which is calculated according to[43]

$$P_V = \frac{4\pi a^3 \sqrt{(1 + 3\cos^2(I))}B}{(2\mu_0)}, \quad (6)$$

where $a = 6371$ km is the radius of the Earth, $\mu_0$ is the permeability of free space and $B$ is in micro Tesla. The rate of change of $P_V$ is then obtained from

$$\frac{\partial P_V}{\partial t} = \frac{P_V(t_1) - P_V(t)}{\Delta t}. \quad (7)$$

**A simple model of rapid directional changes**. We generalise the model in ref. [15] in which an isolated patch is inserted into the radial magnetic field $B_r$ at the CMB, radius $r = c = 3485$ km, centred at the point $\theta_c$, $\phi_c$ in spherical polar coordinates $(r, \theta, \phi)$. The functional form is based on the Fisher–Von Mises probability density function and is given by

$$B_r^s(c, \theta, \phi) = A\kappa \frac{\exp^{\kappa\cos\theta}}{4\pi\sinh\kappa} - \frac{A}{4\pi}, \quad (8)$$

where

$$\cos\theta = \sin\theta\cos\phi\sin\theta_c\cos\phi_c + \sin\theta\sin\phi\sin\theta_c\sin\phi_c + \cos\theta\cos\theta_c \quad (9)$$

and

$$\kappa = \frac{6561}{\sigma_M^2}, \quad (10)$$

where $\sigma_M$ is the standard deviation in both latitude and longitude. The factor $A/4\pi$ ensures that the patch field defined by Eq. (8) remains divergence-free, i.e.

$$\int B_r^s dS_c = 0, \quad (11)$$

where $S_c$ denotes the core surface.

The patch field $B_r^s$ can be expanded in a convergent spherical harmonic series on the unit sphere. It is then straightforward to add to $B_r^s$ a background field $B_r^b$ such that the total radial field $B_r = B_r^s + B_r^b$. Here we set $B_r^b$ equal to an axial dipole field. The total field on the CMB can then be upward continued to the surface where magnetic elements $X$, $Y$, $Z$, intensity $F$, inclination $I$, declination $D$ can be calculated at all points on a global grid.

The aforementioned model produces a stationary field on the CMB. Here we generalise this model to allow the patch to move across the core surface. This is accomplished by introducing two additional parameters that define the latitudinal and longitudinal increments to the central patch location, $\delta\lambda_c$ and $\delta\phi_c$, respectively. Over a sequence of $n = 1, ..., N$ iterations the centre of the patch on the CMB is moved from its original location $(\theta_c, \phi_c)$ to the locations $(\theta_c + n\delta\lambda_c, \phi_c + n\delta\phi_c)$. Of course, there is no physical significance to the increments used in the calculations and hence absolute rates of change cannot be calculated by this approach. Therefore, we simply set $\Delta t = 1$ in Eqs. (2) and (5). We calculate $\partial\hat{\mathbf{B}}/\partial t$ and $\partial\hat{\mathbf{P}}_V/\partial t$ at all locations on a 2° × 2° latitude–longitude grid at Earth's surface and compare values for imposed normal and reversed CMB patches.

The generalised model of an isolated CMB flux patch is completely specified by the patch amplitude $A$, its central location and increments, $\theta_c$, $\delta\lambda_c$, $\phi_c$, $\delta\phi_c$, and width $\sigma_M$. We fix $\sigma_M = 15°$, which produces a broad patch on both the CMB and surface that is easy to identify in visualisations. We set a background axial dipole field with an amplitude of 80 $\mu$T at the CMB, similar to the present geomagnetic

field, and choose $A = -10\,\mu$T and $A = 10\,\mu$T to simulate the two cases of normal and reversed flux patches, respectively. Values of $\theta_c$, $\delta\lambda_c$, $\phi_c$ and $\delta\phi_c$ are varied for the two cases. For the chosen background field, the solutions are identical for positive and negative values of $\delta\phi_c$ and so we focus on the former. We consider both positive and negative values of $\delta\lambda_c$ to simulate patches moving towards and away from the pole, respectively.

**Masking features of the CMB field**. The goal here is to modify the local strength of the radial CMB field in order to evaluate the effect of specific field features on directional changes at the surface. We refer to this process as "masking". Local manipulation of the field must be done in real space; however, the field must also be represented in spherical harmonics in order to perform upward/downward continuation between the surface and CMB. Therefore, any local change to the field in real space will alter the structure globally when represented in spherical harmonics, leading to the possibility of aliasing. To mitigate this effect, we retain all spherical harmonic coefficients in the expansion of the radial field up to the truncation used in the original simulations, usually degree $L = 128 - 256$. For each simulation, sets of high-resolution Gauss coefficients are produced for the 5 time points surrounding an extreme change in the VGP.

The Schmidt-normalised Gauss coefficients $g_l^m$ of degree $l$ and order $m$ at Earth's surface are related to the coefficients $c_l^m$ for the radial field at the CMB by

$$c_l^m = (l+1)g_l^m\left(\frac{a}{c}\right)^{l+1}. \tag{12}$$

The $c_l^m$ are transformed to and from real space using *Shtools*[44] with Gauss–Legendre quadrature and $(3/2)m$ latitude points to ensure an exact transform. In real space, a spatial region is identified and the field there is multiplied by a factor $f$. After experimentation with values of $0.01 \leq f \leq 0.9$ the value $f = 0.5$ was chosen as this is approximately the fractional difference between the maximum and mean absolute field strength in many simulations and also ensures that the field structure remains smooth with a convergent spherical harmonic representation.

In each simulation, we mask four different regions in turn, chosen to be the four $30° \times 30°$ quadrants with a common vertex at the location of maximum change. The quadrants will be referred to as NW (northwest), NE (northeast), SW (southwest) and SE (southeast). For each of the 5 sets of Gauss coefficients (the original plus one each for the 4 quadrants), we compute directional changes as described above, creating maps of $X$, $Y$ and $Z$ on Earth's surface and using these to extract rates of change at all locations.

An example calculation (Supplementary Fig. 7) shows that the method is effective at masking specific regions, as can be clearly seen for the NE quadrant in the middle row. The global influence of masking can be estimated from the power spectrum of the Gauss coefficients at the CMB (Supplementary Fig. 7). This shows little influence from the masking procedure, which is confirmed by visual inspection of global plots of the CMB field (not shown).

## Data availability
All data produced for this manuscript is available from the authors on reasonable request.

## Code availability
The dynamo code used to produce the simulations has been described and benchmarked in ref. [45]. The postprocessing steps that produce grids of magnetic elements and VGPs are standard and are described in refs. [46,13]. The code used to produce rates of change is described in ref. [13] and is available from the authors on reasonable request.

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

## Acknowledgements

C.J.D. acknowledges a Natural Environment Research Council personal fellowship, reference NE/L011328/1. This work was also supported by US National Science Foundation grant EAR1623786 for C.G.C.

## Author contributions

C.J.D. developed the ideas, wrote and ran the computer codes, analysed the results and wrote the manuscript. C.G.C. developed the ideas, analysed the results and wrote the manuscript.

## Competing interests

The authors declare no competing interests.
