## [Peer Review File · Nature Communications]

Reviewers' comments:

Reviewer #1 (Remarks to the Author):

Dear authors and editor,

Authors Davies and Constable present in their work titled "Rapid geomagnetic changes inferred from Earth observations and numerical simulations" the results of multiple numerical geodynamo simulations with a comparison to the recent 100 kyr observational field model GGF100k by Panovska et al. (2018). The simulated models span a range of parameters, however still not reaching the values in the Earth's core, as stated by the authors themselves.

The key results of this work are: a) Based on simulations, GGF100k and Holocene field model CALS10k.2 the maximum directional changes occur more frequently at latitudes $< 40^\circ$ and the maximum intensity changes at latitudes $> 40^\circ$, the latter result already suggested by Davies and Constable (2018) and b) Maximum directional changes are related to reversed flux patches (reversed as in opposite of what is expected on that hemisphere by a dipolar field) and maximum intensity changes are due to normal-polarity flux movement across core surface. However, again, the latter result was already suggested by Davies and Constable (2018).

From a physical point of view these conclusions are plausible. Same polarity flux patch as the general polarity of the surrounding hemisphere would naturally increase the Earth's magnetic field strength at that location, but not necessary create huge directional changes. On the contrary, there, the reversed flux patch would create larger changes in directions, but not create a great spike in the intensity. One could compare it to a very simple experiment on bringing either the N or S pole of a magnet close to N side of a compass needle: the N pole of the magnet will move the N compass needle only slightly, but it will "increase" the strength, the S pole of the magnet will repel the N magnetic needle to 180° from it, but will be slightly smaller in strength than the earlier case. Similarly, extremal directional changes occurring generally at latitudes $< 40^\circ$ and the maximum intensity changes at latitudes $> 40^\circ$ seems feasible. For the Earth's magnetic field both reversed and normal dipolar polarity configurations are equally attractive and on average dipolar field dominates. If this arises simply from a statistically more reversed or normal flux patches distributed on the normal or reversed hemisphere then naturally there would be more crossing over to the wrong polarity on the mid to lower latitudes, but not on higher latitudes.

The paper is well written and provides some new and important information; however, I am not completely convinced of all of its novelty nor the appropriateness for the chosen journal. The work is important, but I feel it might be more suitable for publishing in another journal, perhaps more specifically oriented for geomagnetic and paleomagnetic community.

Davies and Constable state on lines 225-226 that they have added 7 simulations to 9 simulations previously presented by Davies and Constable (2018). However, comparing the text on lines 226-228 and the Supplementary Table 1, it appears to me that from these seven simulations, only three are truly new simulations ($E = 10^{-3}$) and all the rest have been published before. It is therefore at times not completely clear to me what is new/different in this paper compared to the previously published works Davies and Constable (2018), Davies and Constable (2014) and Mound et al. (2015). Furthermore, there are multiple locations where the authors compare their new results with their previous study (Davies and Constable, 2018), and they are in agreement (e.g. lines 97-100, 112-113, 147-149, 163-164). This is natural since more than half of the simulations are the same, but maybe also quite redundant.

As stated earlier, Davies and Constable (2018) already provided support for the intensity spikes with their numerical simulations that gave similar values of dB/dt and spikes arising from the migration of normal-polarity flux patches across the core surface. Therefore, the real novelty that the authors present in the current study is then looking at more detail on the directional changes and doing comparison between rapid directional and intensity variations.

Data quality is good as well as the presentation. However, I wish the authors would make all the figure captions completely self-explanatory and add some text in some figures so that it is faster for the reader to understand. For example, in Figure 1 by writing out what P is as well as Rm. Also adding text above each row: "Non-reversing", "Reversing" and "Observational field model GGF100k" would make the Figure 1 easier and faster for the reader to interpret. The reference for Panovska et al. 2018 should also be mentioned somewhere in the figure caption in all the figures where data from GGF100k is used. In Figures 2, 3 and 4 add what P and B are. And in Figure 5 add what P, E, Pm, Ra. Also add in Figure 5 an arrow or similar which reversed flux is the one responsible for the extremal event.

Some minor suggestions:

Line 14. Rather use "latitudinal ranges" instead of "locations" for example, since there is no longitudinal preference observed. Also is the "extreme field variations" referring to strength, if yes, might be more understandable replacing it with strength instead.

Lines 22 – 23. The result "intensity spikes caused by migration of normal polarity flux patches across the core surface" is not a result of this study, but that of Davies and Constable (2018).

Lines 65 – 75. I feel a lot of this could be moved to methods without jeopardizing the clarity of the work.

Lines 77 – 79. Both of these simulations were already published in Davies and Constable (2018). Is there a reason not to choose an example from the three new simulations with $E = 0.001$?

Line 111. Remove "s" from "Lachamps".

Line 151. Remove extra "." from the end of the sentence.

Supplementary data:

I would suggest to make the figure captions here as well self-explanatory.

In Figure 1 include what P and Rm are. The letters assigned to the Figure 1 should also be checked – there is a duplicate of a, b, c and d.

In Figures 2, 3, 4, 5 and 6 add what P, E, Pm and Ra are. Also adding text Before, During and After above the columns would make these figures faster to interpret. It would be great for the reader if the authors would add a small arrow or similar to point to the suggested causes of the extremal events on the CMB maps. In its current form it is sometimes difficult to understand which flux patch the authors are referring to. Furthermore, all the data in the figures 2 – 6 are from Davies and Constable 2014

Supplementary Figure 4. Could the cause here also be the reversed flux path in southern hemisphere (red) moving from East to West below the location of extremal change?

Supplementary Table 1. Add in some way what simulation is from what publication: Davies and Constable (2014), Davies and Constable (2018) or Mound et al. (2015). Replace "first column" with "first two letters" and "second column" with "last two letters" for clarity when discussing the thermal boundary condition column. Also write out what is B.

Supplementary Table 2. Write out what are Rm, P and B.

References:

Davies, C. & Constable, C. Searching for geomagnetic spikes in numerical dynamo simulations. *Earth Planet. Sci. Lett.* 504, 72–83 (2018).

Davies, C. & Constable, C. Insights from geodynamo simulations into long-term geomagnetic field behaviour. *Earth Planet. Sci. Lett.* 404, 238–249 (2014).

= $E = 5 \times 10^{-4}$ and 1.2×10^{-4} , supp Figs 2, 3, 4,5 and 6

Mound, J., Davies, C. & Silva, L. Inner core translation and the hemispheric balance of the geomagnetic field. *Earth Planet. Sci. Lett.* 424, 148–157 (2015).

= $E = 10^{-5}$

Panovska, S., Constable, C. & Korte, M. Extending global continuous geomagnetic field reconstructions on timescales beyond human civilization. *Geochem. Geophys. Geosys.* (2018).

Reviewer #2 (Remarks to the Author):

This manuscript concerns rapid changes in Earth's magnetic field intensity and direction (spikes). The authors compare paleomagnetic models for the Holocene and for the last 100 ka to a series of numerical simulations, both reversing and non-reversing, over a range of magnetic Reynolds number. The simulations have been developed by the authors and are published and described elsewhere. The paper is well written and referenced. The text is of appropriate length and the figures are complete and legible.

The authors make a series of points after examining the simulations and comparing them to the paleomagnetic models:

1) Extreme direction changes are comparable to GGF100k and Holocene paleomagnetic models but larger than present day rates of change for field direction and intensity.

2) Extreme intensity changes in the simulations are larger than in these same models (but in line with the paleomagnetic examples such as the Levantine intensity spike)

3) Most rapid directional changes occur at low paleolatitudes (< 40 degrees) while major intensity changes occur at high latitudes (>40 degrees)

4) Fast directional changes are not coincident with axial dipole reversals. Reversing simulations show a lag of up to 5 ka. The authors mention that directional spikes may be a precursory phase to a reversal.

5) Extreme directional and intensity events appear to have different physical origins. They do not occur together and in fact show a latitudinal separation with each type tending to occur in different regions.

6) Extreme directional events appear associated with reversed flux patches at the CMB moving toward the extreme event location. Extreme intensity events appear associated with migration of normal flux patches

7) These results, as interpreted by the authors, suggest a linkage between core processes and paleomagnetic observations.

Taken as a whole, the authors present a plausible set of interpretations that are consistent between the paleomagnetic record and their simulations. There are concerns, however, regarding the robustness of these interpretations. Some of the interpretations (for example low/high latitude assertions) could be better supported with statistics showing their significance. There is also a question of uniqueness. Showing, for example, the CMB magnetic field at three snapshots in time (Figure 5) reveal many areas that are changing considerably yet the authors pick out specific

observations (e.g. a reversed flux patch heading toward the excursion event) as an explanation for the rapid changes in surface field direction or intensity. These are of course highly complex numerical simulations – given this I find the interpretation to be rather subjective and speculative.

Along a similar vein, I find the discussion about field reversals and their relation to direction and/or intensity spikes to be not well supported. Beyond saying that there is little correlation I don't see the statement about spikes as a potential precursor to a field reversal as having much support. Is it not just as likely (or more likely) that field reversals have absolutely no causal association with these rapid spikes in intensity/direction? Without delving into some statistics here I think the relation between reversals and spikes is unwarranted.

More broadly, I wonder whether this work is of sufficient interest and significance to a broader scientific audience. Certainly a new understanding of magnetic field reversals meets these criteria as would demonstrating a convincing linkage between core processes and paleomagnetic spikes. Though the authors mention that latter, as written I don't see that this manuscript shows this. It does advance our understanding of the magnitude of extreme directional and intensity changes, but I wouldn't say it lays to rest the controversy about intensity spikes or rapid directional changes in the paleomagnetic record. Finally, to quote the authors initial sentence, it is not clear how the understanding gained from these numerical simulations are 'crucial to predicting future field changes.'

Finally, as written, this manuscript covers a lot of points and it is challenging to see a single coherent story within it. Of the 7 points that I took away from this work, I find that #5 (that extreme directional and intensity deviations appear unrelated) is the most relevant result here. This result is, however, of more limited interest to the paleomagnetic community than perhaps the broader scientific community.

Reviewer #3 (Remarks to the Author):

In a previous paper (EPSL, 2018), Davies and Constable studied whether numerical simulations of the geodynamo show sudden changes in the intensity of the Earth's magnetic field at some geographical location. Their study was motivated by the rapid intensity variations suspected from analyses of archeomagnetic artefacts from the Levant and dated 3000 years BP. Here, the authors complement this first study with a similar analysis for the local direction of the magnetic field. Rapid changes in the magnetic field direction characterize magnetic reversals and excursions. They rely on some of their previous works to scale time and magnetic field intensity in the simulations.

I find the question addressed in the paper worthy of attention. The paper is potentially very interesting. However, I think that the statistical analysis needs to be much improved. I have indeed concerns about joint analysis of rapid changes in the intensity and in the direction of the Earth's magnetic field. The type of intensity spikes Davies and Constable investigate may be much more frequent than the rapid angle variation they discuss. The authors motivate their work by a spike that may have occurred 3000 years ago while the rapid direction changes they mention may have happened either during the last reversal 780 000 years ago or during the Laschamp excursion 40 000 years ago. Thus, I find it necessary that the authors show histograms of rapid events (intensity and direction changes) and that they discuss possible conflicts (in terms of frequency of occurrence) with our current views about the Earth's magnetic field. It would be nice to have histograms for different latitude bands.

I have also concerns about the magnetic field model GGF100k on which they rely to discuss the evolution of the Earth's magnetic field. As I understand it, it covers only one large and global excursion, the Laschamp one. As a result, I suspect it would be much useful to rely also on specific analyses of this event, such as Leonhardt & al. (EPSL, 2009, 278, 87-95) and Ingham & al. (EPSL,

2017, 472, 131-141). The model GGF100k is tuned to describe the field over 100 000 years; it will be useful to calculate histograms but less so to give the detailed history of the Laschamp event (see the numerous data rejected around the Laschamp excursion to build the model according to Panovska & al., 2018).

Other comments:

The authors explain that intensity spikes mainly occur at latitudes larger than 40 degree whereas the latitude of Jerusalem (Levant) is about 30 degree. Is there a contradiction here?

Figure 1, bottom: Is it correct that you rely here only on one time series (sediment record) to characterize the most rapid changes during the Laschamp event ? It is all the more annoying that a lot of data around the Laschamp excursion were discarded to construct GGF100k.

Revisions to NCOMMS-19-29765, “Rapid geomagnetic changes inferred from Earth observations and numerical simulations”.

We are very grateful for the detailed and constructive comments provided by all three reviewers. The main changes to the manuscript address the following points:

- 1) Novel contribution. We have removed intensity variations from the results section to clearly show that the major novelty of the work is new predictions of and physical insight into rapid directional changes of the geomagnetic field. This change does not reduce the novelty of our study since all conclusions regarding intensity spikes were published in Davies and Constable (2018, hereafter DC18). Consequently Figure 4 of the original manuscript has been removed, Figures 2 and 3 have been combined with intensity results removed, and intensity results in Figure 1 and Supplementary Table 2 have been removed. We still compare rapid intensity and directional changes in the Discussion section, but now with reference to the intensity results in DC18. The novelty and advance provided by our study have also been enhanced by the work conducted to address points 2 and 3 below.
- 2) Statistics. We have computed histograms, PDFs and CDFs for all of our simulations and also the global field models GGF100K and LSMOD.2. From this we show that directional changes are approximately log-normally distributed and that the fastest changes occur at lower latitudes. This latter conclusion was obtained in the original manuscript when only considering extremal events. These results are shown in the new Figure 3.
- 3) Links to core processes. We have developed two new tools that provide complementary insight into the core processes that cause rapid directional changes: 1) a simple model of the CMB field comprising a single normal/reversed moving flux patch superimposed on an axial dipole field; 2) a method for locally masking features of the CMB field in dynamo simulations. Applying both tools to our dynamo simulations reinforces the importance of reversed flux patch migration in producing rapid directional changes. These results are shown in the new Figures 5 and 6. Plots showing the local CMB field surrounding extreme events and graphs showing the contributions of different regions of CMB field to the observed rates of change have been added to Supplementary Figures 2-6.

This work has resulted in significant revisions to the manuscript as described in detail below. Despite this the main conclusions of the original paper remain unchanged and, we believe, have been strengthened by a more detailed analysis of directional changes alone.

In the following text reviewer comments are reproduced in black with responses in blue. All line and figure numbers in responses refer to the revised manuscript unless explicitly stated.

Reviewer #1 (Remarks to the authors)

Authors Davies and Constable present in their work titled “Rapid geomagnetic changes inferred from Earth observations and numerical simulations” the results of multiple numerical geodynamo simulations with a comparison to the recent 100 kyr observational field model GGF100k by Panovska et al. (2018). The simulated models span a range of parameters, however still not reaching the values in the Earth’s core, as stated by the authors themselves.

1. The key results of this work are: a) Based on simulations, GGF100k and Holocene field model CALS10k.2 the maximum directional changes occur more frequently at latitudes $< 40^\circ$ and the maximum intensity changes at latitudes $> 40^\circ$, the latter result already suggested by Davies and Constable (2018) and b) Maximum directional changes are related to reversed flux patches (reversed as in opposite of what is expected on that hemisphere by a dipolar field) and maximum intensity changes are due to normal-polarity flux movement across core surface. However, again, the result was already suggested by Davies and Constable (2018)

We agree with the referee that the intensity results discussed in the original manuscript had essentially been obtained in DC18 and therefore somewhat obscured the novel contributions of the present work. To be clear, DC18 investigated the properties of intensity spikes whereas the present paper is focused on directional changes. To address this point we have made a significant change to the manuscript by removing intensities from the results section entirely. This change does not reduce the novelty of the paper because, as the reviewer points out, the intensity results were simply covering old ground, and has the added benefit of leaving more space to address the statistics and physical origin of rapid directional changes (in response to comments by Reviewers 2 and 3 below). Intensity variations are still compared to directional changes in the Discussion section, but now this is done with reference to the results in Davies and Constable (2018) with no loss of generality. Specific changes to the manuscript are described in response to comment #5 below.

Document Changes: Removed intensity variations from the results section.

2. From a physical point of view these conclusions are plausible. Same polarity flux patch as the general polarity of the surrounding hemisphere would naturally increase the Earth’s magnetic field strength at that location, but not necessary create huge directional changes. On the contrary, there, the reversed flux patch would create larger changes in directions, but not create a great spike in the intensity. One could compare it to a very simple experiment on bringing either the N or S pole of a magnet close to N side of a compass needle: the N pole of the magnet will move the N compass needle only slightly, but it will “increase” the strength, the S pole of the magnet will repel the N magnetic needle to 180° from it, but will be slightly smaller in strength than the earlier case. Similarly, extremal directional changes occurring generally at latitudes $< 40^\circ$ and the maximum intensity changes at latitudes $> 40^\circ$ seems feasible. For the Earth’s magnetic field both reversed and normal dipolar polarity configurations are equally attractive and on average dipolar field dominates. If this arises simply from a statistically more reversed or normal flux patches distributed on the normal or reversed hemisphere then naturally there would be more crossing over to the wrong polarity on the mid to lower latitudes, but not on higher latitudes.

We are grateful for the reviewer’s thoughts on the physical origin of rapid field changes. Based on this and the comment of Reviewer 2 we have developed a simple model of the CMB field

that combines a single normal/reversed moving flux patch with a static axial dipole. This model shows that motion of reversed patches across the core surface always leads to faster directional changes than an equivalent normal polarity patch. It further suggests that the fastest changes occur in regions where opposite signed flux from the patch and dipole field balance, leaving a region where changes are dictated by smaller scale (and hence more rapidly time-varying) field components.

Document Changes: The simple model is described in the new section of the revised methods entitled “A simple model of rapid directional changes” and the new section of the main text entitled “Linking rapid directional changes and core processes”.

3. The paper is well written and provides some new and important information; however, I am not completely convinced of all of its novelty nor the appropriateness for the chosen journal. The work is important, but I feel it might be more suitable for publishing in another journal, perhaps more specifically oriented for geomagnetic and paleomagnetic community.

We hope that the three major changes to the manuscript explained at the start of this letter have clarified the novelty of our work.

We believe that our paper is suitable for publication in Nature Communications for the following reasons. First, the subject has gained a lot of recent attention from both observational and modelling perspectives with several papers published in Letter-style journals (including our previous work on intensity spikes that was published in Nature Communications and the LSMOD.1 model of the Laschamp excursion published in PNAS). Second, the paper exploits the recent development of the GGF100k model, the first and only global time-varying model of Earth’s magnetic field spanning the last 100 kyr. This model has a broad range of applications, e.g. to core dynamics, geomagnetism, geochronology, atmospheric chemistry (cosmogenic isotope production), space climate, and its use here will be of interest to researchers in these areas. Third, we believe that the work is of broad interest to communities beyond paleomagnetism because it relies on a synthesis of results from dynamical simulations and geo/paleomagnetism in order to provide new insight into the dynamics of Earth’s core and the geomagnetic field. Indeed, we believe that the excellent agreement between these completely independent probes of geomagnetism demonstrates the significant advances and growing maturity of these fields over the last 20 years. In the revised manuscript we have added a final paragraph that discusses the broader implications of our work and potential future directions.

Document changes: Added paragraphs on L53-61 and L239-254.

4. Davies and Constable state on lines 225-226 that they have added 7 simulations to 9 simulations previously presented by Davies and Constable (2018). However, comparing the text on lines 226-228 and the Supplementary Table 1, it appears to me that from these seven simulations, only three are truly new simulations ($E = 10^{-3}$) and all the rest have been published before. It is therefore at times not completely clear to me what is new/different in this paper compared to the previously published works Davies and Constable (2018), Davies and Constable (2014) and Mound et al. (2015).

The reviewer is correct that none of the simulations are new in the sense that they were conducted specifically for this work. Indeed, the simulations were chosen because they have

been previously studied in a different context and are therefore relatively well understood. However, the analysis we conduct is entirely new.

Document Changes: L269-273. We explain why the simulations were selected on L64-65.

5. Furthermore, there are multiple locations where the authors compare their new results with their previous study (Davies and Constable, 2018), and they are in agreement (e.g. lines 97-100, 112-113, 147-149, 163-164). This is natural since more than half of the simulations are the same, but maybe also quite redundant.

We agree that this information is redundant.

Document Changes: removed the comparisons on L97-98 and L112-113; removed the paragraph on L123-130 that compared intensity and directional spikes and the associated figure 4; removed the statements on L163-164. We have retained the statements on L147-149 (now L186-188) because they justify our choice of simulations.

6. As stated earlier, Davies and Constable (2018) already provided support for the intensity spikes with their numerical simulations that gave similar values of dB/dt and spikes arising from the migration of normal-polarity flux patches across the core surface. Therefore, the real novelty that the authors present in the current study is then looking at more detail on the directional changes and doing comparison between rapid directional and intensity variations.

The reviewer is correct. As explained in response to comment #1 above, we have now focused the results on directional changes only so that the novelty of the work is clear. Intensity and directional changes are compared in the discussion using the previous results of DC18.

Document Changes: Lines 225-238.

7. Data quality is good as well as the presentation. However, I wish the authors would make all the figure captions completely self-explanatory and add some text in some figures so that it is faster for the reader to understand. For example, in Figure 1 by writing out what P is as well as Rm. Also adding text above each row: “Non-reversing”, “Reversing” and “Observational field model GGF100k” would make the Figure 1 easier and faster for the reader to interpret. The reference for Panovska et al. 2018 should also be mentioned somewhere in the figure caption in all the figures where data from GGF100k is used. In Figures 2, 3 and 4 add what P and B are. And in Figure 5 add what P, E, Pm, Ra. Also add in Figure 5 an arrow or similar which reversed flux is the one responsible for the extremal event.

We have made the changes suggested by the reviewer.

Some minor suggestions:

Line 14. Rather use “latitudinal ranges” instead of “locations” for example, since there is no longitudinal preference observed. Also is the “extreme field variations” referring to strength, if yes, might be more understandable replacing it with strength instead.

We have made the suggested change regarding “latitude ranges”. The statement regarding “extreme field variations” has been removed due to the major revision point (1) above.

Lines 22 – 23. The result “intensity spikes caused by migration of normal polarity flux patches across the core surface” is not a result of this study, but that of Davies and Constable (2018).

This is correct. This statement has been removed from the revised abstract.

Lines 65 – 75. I feel a lot of this could be moved to methods without jeopardizing the clarity of the work.

We have decided to retain this text as it helps to introduce the notation and terminology that is used throughout the section.

Lines 77 – 79. Both of these simulations were already published in Davies and Constable (2018). Is there a reason not to choose an example from the three new simulations with $E = 0.001$?

We chose these simulations simply because one reverses and one does not, yet both show rapid directional changes. Results from all simulations are shown in Supplementary Figure 1.

Line 111. Remove “s” from “Laschamps”.

We have made the suggested change.

Line 151. Remove extra “.” from the end of the sentence.

Done.

Supplementary data:

I would suggest to make the figure captions here as well self-explanatory.

We have made changes analogous to those suggested by the reviewer in comment #7 above.

In Figure 1 include what P and R_m are. The letters assigned to the Figure 1 should also be checked – there is a duplicate of a, b, c and d.

We have made the suggested changes and thank the reviewer for noticing the issue with the figure labels.

In Figures 2, 3, 4, 5 and 6 add what P, E, P_m and R_a are. Also adding text Before, During and After above the columns would make these figures faster to interpret. It would be great for the reader if the authors would add a small arrow or similar to point to the suggested causes of the extremal events on the CMB maps. In its current form it is sometimes difficult to understand which flux patch the authors are referring to. Furthermore, all the data in the figures 2 – 6 are from Davies and Constable 2014

We thank the reviewer for these excellent suggestions and have made the required changes. We do note however that while the simulations in figures 2-6 were first published in Davies and Constable (2014), that study focused on long-term evolution of global properties of the field. The present study is the first to analyse rapid and localised field changes in these

simulations, which required significant post processing of the “raw” simulation outputs (these being spherical harmonic representations of the toroidal and poloidal scalar fields within the dynamo region). We therefore consider that all of the data in figures 2-6 is new.

Supplementary Figure 4. Could the cause here also be the reversed flux path in southern hemisphere (red) moving from East to West below the location of extremal change?

The reviewer is absolutely correct and we now show this quantitatively in the revised Supplementary Figure 4 by masking different regions of the CMB field (see main point 1 above).

Supplementary Table 1. Add in some way what simulation is from what publication: Davies and Constable (2014), Davies and Constable (2018) or Mound et al. (2015). Replace “first column” with “first two letters” and “second column” with “last two letters” for clarity when discussing the thermal boundary condition column. Also write out what is B.

We have made the suggested changes.

Supplementary Table 2. Write out what are R_m , P and B .

We have made the suggested changes.

Reviewer #2 (Remarks to the Author):

This manuscript concerns rapid changes in Earth's magnetic field intensity and direction (spikes). The authors compare paleomagnetic models for the Holocene and for the last 100 ka to a series of numerical simulations, both reversing and non-reversing, over a range of magnetic Reynolds number. The simulations have been developed by the authors and are published and described elsewhere. The paper is well written and referenced. The text is of appropriate length and the figures are complete and legible.

The authors make a series of points after examining the simulations and comparing them to the paleomagnetic models:

- 1) Extreme direction changes are comparable to GGF100k and Holocene paleomagnetic models but larger than present day rates of change for field direction and intensity.
- 2) Extreme intensity changes in the simulations are larger than in these same models (but in line with the paleomagnetic examples such as the Levantine intensity spike)
- 3) Most rapid directional changes occur at low paleolatitudes (< 40 degrees) while major intensity changes occur at high latitudes (>40 degrees)
- 4) Fast directional changes are not coincident with axial dipole reversals. Reversing simulations show a lag of up to 5 ka. The authors mention that directional spikes may be a precursory phase to a reversal.
- 5) Extreme directional and intensity events appear to have different physical origins. They do not occur together and in fact show a latitudinal separation with each type tending to occur in different regions.
- 6) Extreme directional events appear associated with reversed flux patches at the CMB moving toward the extreme event location. Extreme intensity events appear associated with migration of normal flux
- 7) These results, as interpreted by the authors, suggest a linkage between core processes and paleomagnetic observations.

We note that, based on the comments of all reviewers, point 2 no longer appears in the manuscript and point 4 has been toned down (no mention of lag times or a potential precursory signal to reversals), while points 5 and 6 are mentioned in the discussion as implications of our work. The revised manuscript therefore contains three main conclusions corresponding to points 1, 3 and 7 above, which we hope has simplified the presentation.

1. Taken as a whole, the authors present a plausible set of interpretations that are consistent between the paleomagnetic record and their simulations. There are concerns, however, regarding the robustness of these interpretations. Some of the interpretations (for example low/high latitude assertions) could be better supported with statistics showing their significance.

We agree with the reviewer that our on focus extremal events was not sufficient to provide a robust statistical description of the latitude-dependence of rapid field changes. To address this

issue we have now computed histograms of directional changes for all simulations and the GGF100k model. In the new Figure 3 we demonstrate that histograms of rates of change are well represented by lognormal distributions both globally and at specific latitudes. We have computed probability and cumulative distribution functions at different latitudes for all simulations and GGF100k, which clearly show the greater probability of observing faster directional changes at lower latitudes.

Document Changes: New Figure 3 and accompanying text on L116-132 and L208-212.

2. There is also a question of uniqueness. Showing, for example, the CMB magnetic field at three snapshots in time (Figure 5) reveal many areas that are changing considerably yet the authors pick out specific observations (e.g. a reversed flux patch heading toward the excursion event) as an explanation for the rapid changes in surface field direction or intensity. These are of course highly complex numerical simulations – given this I find the interpretation to be rather subjective and speculative.

This is another fair criticism and an important point. To address it we have developed two new tools that provide complementary insight into the core processes that cause rapid directional changes: 1) a simple model of the CMB field comprising a single normal/reversed moving flux patch superimposed on an axial dipole field; 2) a method for locally reducing the strength of the radial CMB field in dynamo simulations. The resulting calculations reinforce the importance of reversed flux patch migration in producing rapid directional changes. Moreover, these tools have allowed us to isolate the specific features of the CMB field that produced extreme directional changes in the simulations that we were unable to interpret in the original manuscript (Supplementary Figures 2, 4, 5, of the original manuscript). We are therefore extremely grateful to the reviewer for encouraging us to improve this aspect of the work; it has made a significant improvement to the paper.

Document Changes: We have created a new section entitled “Linking rapid directional changes and core processes” in the main text, which references two new figures: Figure 4 synthesises the results from the simple model while Figure 6 shows results from the masking. The two tools are described in new Methods sections entitled “A simple model of rapid directional changes” and “Masking of CMB field features”. Discussion is added on L218-224.

3. Along a similar vein, I find the discussion about field reversals and their relation to direction and/or intensity spikes to be not well supported. Beyond saying that there is little correlation I don't see the statement about spikes as a potential precursor to a field reversal as having much support. Is it not just as likely (or more likely) that field reversals have absolutely no causal association with these rapid spikes in intensity/direction? Without delving into some statistics here I think the relation between reversals and spikes is unwarranted.

We agree that this observation was not well justified in the original manuscript. It could be addressed by identifying periods of stable and transitional polarity in the simulations and applying the statistical analysis (see main point 2 above) to the two different periods independently, but we have not attempted this. Instead we have removed this text from the discussion section and abstract.

Document Changes: Removed text on relations between rapid directional changes and reversals from the abstract and discussion.

5. More broadly, I wonder whether this work is of sufficient interest and significance to a broader scientific audience. Certainly a new understanding of magnetic field reversals meets these criteria as would demonstrating a convincing linkage between core processes and paleomagnetic spikes. Though the authors mention that latter, as written I don't see that this manuscript shows this. It does advance our understanding of the magnitude of extreme directional and intensity changes, but I wouldn't say it lays to rest the controversy about intensity spikes or rapid directional changes in the paleomagnetic record. Finally, to quote the authors initial sentence, it is not clear how the understanding gained from these numerical simulations are 'crucial to predicting future field changes.'

We have undertaken significant revisions (described above) that we believe convincingly link rapid directional changes to core processes. Of course we still cannot claim that our work ends existing controversies surrounding rapid changes, which also requires efforts in paleomagnetism, geochronology, and rock magnetism. Such progress will only come from a large body of evidence and we believe our work provides a new perspective that can aid the detection and interrogation of rapid changes.

Regarding the final point, a rapid field change in unit time that is not correctly predicted will lead to a greater misfit to the 'true' solution than a slow change, all other factors being equal. It is already well known that geomagnetic jerks hinder predictions based on the observed secular variation and we might expect a similar issue to arise with rapid directional changes. The purpose of this sentence was not to claim that our work would improve predictions of future field behaviour, but rather that characterising such behaviour and providing a physical basis for it would be a step towards this broader goal. Nevertheless, to avoid any potential misunderstanding we have removed this sentence from the abstract.

6. Finally, as written, this manuscript covers a lot of points and it is challenging to see a single coherent story within it. Of the 7 points that I took away from this work, I find that #5 (that extreme directional and intensity deviations appear unrelated) is the most relevant result here. This result is, however, of more limited interest to the paleomagnetic community than perhaps the broader scientific community.

We accept that there were a number of conclusions drawn in the original manuscript and the order of importance was not obvious. This was partly because no previous work has (to our knowledge) studied the dynamics of rapid directional changes and hence there are lots of new results we considered to be of potential interest. However, we do believe that the paper presented a coherent story that simulations and field models predict faster directional changes than have ever been observed by paleomagnetic work and that these variations may reflect a distinct physical process in the core. We have rewritten parts of the revised manuscript to more clearly articulate the main message. We also believe that removing intensity variations from the results simplifies the story and provides a much more focused presentation.

Document Changes: Rewritten parts of the abstract and added summary text on L53-61.

Finally, we consider all conclusions drawn in the original manuscript (not just point #5) to be relevant to the results presented. We believe that the additional work arising from the comments of all three reviewers has yielded robust conclusions pertaining to the rates and locations of rapid directional changes as well as their relation to core processes. We still compare rapid directional and intensity changes in the discussion using results from DC18.

Reviewer #3 (Remarks to the Author):

In a previous paper (EPSL, 2018), Davies and Constable studied whether numerical simulations of the geodynamo show sudden changes in the intensity of the Earth's magnetic field at some geographical location. Their study was motivated by the rapid intensity variations suspected from analyses of archeomagnetic artefacts from the Levant and dated 3000 years BP. Here, the authors complement this first study with a similar analysis for the local direction of the magnetic field. Rapid changes in the magnetic field direction characterize magnetic reversals and excursions. They rely on some of their previous works to scale time and magnetic field intensity in the simulations.

I find the question addressed in the paper worthy of attention. The paper is potentially very interesting. However, I think that the statistical analysis needs to be much improved. I have indeed concerns about joint analysis of rapid changes in the intensity and in the direction of the Earth's magnetic field. The type of intensity spikes Davies and Constable investigate may be much more frequent than the rapid angle variation they discuss. The authors motivate their work by a spike that may have occurred 3000 years ago while the rapid direction changes they mention may have happened either during the last reversal 780 000 years ago or during the Laschamp excursion 40 000 years ago. Thus, I find it necessary that the authors show histograms of rapid events (intensity and direction changes) and that they discuss possible conflicts (in terms of frequency of occurrence) with our current views about the Earth's magnetic field. It would be nice to have histograms for different latitude bands.

We thank the referee for suggesting a statistical analysis of rapid field changes, which was not possible in the original manuscript given our decision to focus on extremal events. We have therefore computed histograms, PDFs and CDFs for all simulations as well as the GGF100K and LSMOD.2 global field models. These properties have been computed at different latitudes as suggested by the referee. Note that since intensity variations have been removed from the results in response to comments from Reviewers 1 and 2 we do not compute the statistical properties of the field strength. Instead, when comparing variations in field strength and direction in the discussion we note the concerns expressed by the reviewer. Document Changes: New Figure 3 and accompanying text on L118-132, L195-199, L208-212 and L225-238.

I have also concerns about the magnetic field model GGF100k on which they rely to discuss the evolution of the Earth's magnetic field. As I understand it, it covers only one large and global excursion, the Laschamp one. As a result, I suspect it would be much useful to rely also on specific analyses of this event, such as Leonhardt & al. (EPSL, 2009, 278, 87-95) and Ingham & al. (EPSL, 2017, 472, 131-141). The model GGF100k is tuned to describe the field over 100 000 years; it will be useful to calculate histograms but less so to give the detailed history of the Laschamp event (see the numerous data rejected around the Laschamp excursion to build the model according to Panovska & al., 2018).

We thank the reviewer for raising the issue of resolution in the GGF100k model. The data used by Leonhardt et al., and Ingham et al. are a subset of those used in the GGF100k model analysed here, and their models are now considered to be superseded. Instead, we have added the LSMOD models to our analyses, which provide the most robust global view of the time interval surrounding the Laschamp excursion. They show that the analysis of GGF100k

is limited in resolution so that estimates of rates of change are conservative, but do not alter the main conclusions. GGF100k, however provides the longer term perspective illustrating the comparative rarity of very rapid directional changes.

Document Changes: L79-86, L205-206.

Other comments:

The authors explain that intensity spikes mainly occur at latitudes larger than 40 degree whereas the latitude of Jerusalem (Levant) is about 30 degree. Is there a contradiction here?

We don't believe so. This may well be because the Levantine event remains incompletely defined because of limited spatial and temporal sampling. In any case, we do not discuss this point since the revised manuscript is now focused solely on directional changes (see main point 1 above)

Figure 1, bottom: Is it correct that you rely here only on one time series (sediment record) to characterize the most rapid changes during the Laschamp event? It is all the more annoying that a lot of data around the Laschamp excursion were discarded to construct GGF100k.

The time-series presented in Figure 1 is from the location of the most extreme rate of change globally. As outlined in the methods section, far from relying on a single sediment record, we have used the entire model, sampling at 100 year intervals and every 2 degrees globally, with geographic variations in extremal values plotted in the left panel.

The discarding of some data around Laschamp in creating GGF100k reflects some incompatibility across the various records. Detailed analyses of these incompatibilities, and rationales for merging or discarding of individual records is provided by Brown et al (2019) and Korte et al (2019), in refs 28 and 30. Our further analyses of LSMOD.2 provide more detailed support for the GGF100k results.

REVIEWER COMMENTS

Reviewer #2 (Remarks to the Author):

Review of revised version of 'Rapid geomagnetic changes inferred from Earth observations and numerical simulations' by Davies and Constable.

The previous version of this manuscript suffered from a lack of focus and confusion between past results related to extreme intensity variations and new results on extreme directional changes. The new manuscript is considerably more focused in this regard and reiterates the take-home message more clearly and frequently.

A primary concern of the original manuscript was the rather tenuous assertion that extreme directional changes are attributed to the migration of reversed flux patches at the CMB. As the connection between directional changes and core processes is a primary finding of this paper, this seemingly subjective and non-unique assertion was problematic. The authors have done an impressive amount of work to address this concern, developing machinery to simulate the migration of an isolated normal or reversed flux patch across the CMB. Their work now demonstrates that when considering isolated flux patches, reversed patches produce faster directional changes than normal patches. And as the authors point out, these simulations further hint at the underlying core processes responsible for extreme directional changes.

The authors addressed another major concern through the inclusion of statistics for dPv/dt in the GGF100k model (Fig 3a,b) and the comparison to the simulations (Fig 3c,d). These statistics provide a more solid basis and convincing support to the authors' assertion that the maximum directional changes tend to occur at lower latitudes (<40 degrees).

The considerable revisions to this manuscript have addressed the major shortcomings of the original and strengthened the assertions (1) linking numerical simulations to GGF100k, (2) relating maximum directional changes to low latitudes and (3) attributing them to reverse-polarity flux patches at the CMB. I commend the authors for undertaking the necessary work to accomplish this result. In my opinion, the authors have turned this study into a defensible work that is of sufficient interest and importance to be published in Nature Communications.

Reviewer #3 (Remarks to the Author):

The paper puts great emphasis on two results:

- directional changes of the Earth's magnetic field reach $10^\circ/\text{yr}$ even in times of stable polarity
- these extreme changes are caused by motion of reverse flux patches at the core-mantle boundary

However, I find that the message is not yet clear enough. At the end, I don't know whether the authors think rapid directional changes may occur independently of magnetic excursions. The paper does not include examples of paleomagnetic records of fast changes (up to $10^\circ/\text{yr}$) independently of reversal or excursion of the field. The only geophysical cases they mention are the Matuyama-Brunhes reversal and the Laschamp excursion.

I. 203 we read 'faster directional changes than have been previously considered viable'. Do the authors mean during reversals and excursions? Otherwise, they have no observational evidence.

The authors analysed one magnetic field model GGF100k whereas they refer to two other models LSMOD.1 and LSMOD.2 to support their view that high rates occurred close to the Laschamp

event. Why didn't they analyse these models instead? In my initial report, I commented on the Figure 1, bottom, obtained from the model GGF100k but the authors did not answer my question, which was perhaps not clear enough. I reiterate it. The figure shows a very localized region where the dipole direction changes rapidly. It seems associated with the location of a unique sediment relative paleointensity record (see figure 1a of Panovska & al., 2019). Does the conclusion relies on this record only? How do the authors explain that the figure differs so strikingly from the figures illustrating the numerical simulations?

If we accept that we have no observational evidence of very rapid changes (on the order of $10^\circ/\text{yr}$) not associated with reversals or excursions, do the authors mean that we should expect that such changes occur, on the basis of their numerical simulations? I have several questions related to this point:

- in the abstract: directional change reach 10 deg. per year even in times of stable polarity: can you illustrate this point ?
- l.104-106: large values are not necessarily associated with low values of λ_D , can you point examples to the reader ?
- l 229-230: 'These results suggest that extremal intensity and directional variations reflect different physical processes at the top of the core.' Is it correct to conclude also that extremal directional variations are much rarer than extremal intensity events ?

Supplementary figure 1: explain how you have selected the 20 thousand years period that is shown and give the total duration of the simulation.

Revisions to NCOMMS-19-29765A, “Rapid geomagnetic changes inferred from Earth observations and numerical simulations”.

We are grateful for the constructive comments provided by both reviewers. In the response below, reviewer comments are in black and responses are in blue.

Reviewer #2 (Remarks to the Author):

The considerable revisions to this manuscript have addressed the major shortcomings of the original and strengthened the assertions (1) linking numerical simulations to GGF100k, (2) relating maximum directional changes to low latitudes and (3) attributing them to reverse-polarity flux patches at the CMB. I commend the authors for undertaking the necessary work to accomplish this result. In my opinion, the authors have turned this study into a defensible work that is of sufficient interest and importance to be published in Nature Communications.

We thank the reviewer for their previous comments and are glad they appreciate the significant amount of work undertaken to improve the paper.

Reviewer #3 (Remarks to the Author):

The paper puts great emphasis on two results:

- directional changes of the Earth’s magnetic field reach $10^\circ/\text{yr}$ even in times of stable polarity
- these extreme changes are caused by motion of reverse flux patches at the core-mantle boundary

However, I find that the message is not yet clear enough. At the end, I don’t know whether the authors think rapid directional changes may occur independently of magnetic excursions. The paper does not include examples of paleomagnetic records of fast changes (up to $10^\circ/\text{yr}$) independently of reversal or excursion of the field. The only geophysical cases they mention are the Matuyama-Brunhes reversal and the Laschamp excursion.

We agree that the relation between rapid changes and excursions was not directly confronted in the manuscript. Our main point was to demonstrate that rapid changes of $\sim 10^\circ$ can arise at all since this is much faster than any accepted directional change on record. The best evidence for rapid directional changes in paleomagnetic records and time varying paleofield models does indeed come from times that are close to known reversals and excursions. Our results suggest that rapid changes are not necessarily associated with large deviations of the dipole axis, but are often associated with a decrease in field strength that accompanies excursions. However, we do not yet rule out the possibility that large directional changes might occur during stable polarity times due to rapid growth of a reverse flux patch that might not lead to global large-scale decrease in intensity. Such a scenario is seen in a number of our dynamo simulations (e.g. Supplementary Figure 1e, g, h, k) and may correspond to a so-called “regional” rather than “global” excursion. The quality of the current paleomagnetic record is not adequate to map such regional behaviour in detail, but recent progress in global field modelling suggests it can be achieved in the future.

Document changes: Removed the remark “even in times of stable polarity” from the abstract. Added the information above to the main text on lines 224-233.

I. 203 we read ‘faster directional changes than have been previously considered viable’. Do the authors mean during reversals and excursions? Otherwise, they have no observational evidence.

This point has been addressed under the previous comment.

The authors analysed one magnetic field model GGF100k whereas they refer to two other models LSMOD.1 and LSMOD.2 to support their view that high rates occurred close to the Laschamp event. Why didn't they analyse these models instead?

We did analyse both LSMOD.1 and LSMOD.2 models but chose to focus on the GGF100k results in the paper because the longer temporal context provides a more stable result, albeit of lower overall temporal resolution. We consider LSMOD.2, which supersedes LSMOD.1 the more reliable of the two and had quoted those results on lines 203-205 and 214. We have now added Supplementary Figure 8 (reproduced below), which shows the information in Figure 1 of the main text for LSMOD.2.

Supplementary Figure 8: Rapid directional changes in the observational field model LSMOD.2. Top shows a Mollweide projection at Earth's surface of the largest change in VGP position as a function of location. The red star shows the location of the extreme event. Bottom panel shows directional data at the location of the extreme event over a 20 kyr period with the extreme event at the midpoint.

In my initial report, I commented on the Figure 1, bottom, obtained from the model GGF100k but the authors did not answer my question, which was perhaps not clear enough. I reiterate it. The figure shows a very localized region where the dipole direction changes rapidly. It seems associated with the location of a unique sediment relative paleointensity record (see figure 1a of Panovska & al., 2019). Does the conclusion relies on this record only?

We confess to having been initially confused by the reference to Panovska et al. (2019), and believe the reference is intended to be to the geographical distribution of sites in Fig 1a of Panovska et al. (2018a) shown below. The GGF100k model is the product of the entire data distribution shown in this figure and documented in the supplementary material provided in Panovska et al. (2018b). As can be inferred from part (c) of this figure the global field model is sensitive to widely distributed data, and not uniquely controlled by the Equatorial Pacific record. The geographic distribution of data is uneven,

but the regularization imposed by modelling algorithm is intended to suppress large changes in areas with relatively sparse coverage.

Document changes: We have added sentences in lines 84-89 elaborating on the differences between the simulations and GGF100k and noting that GGF100k has broad support from global data and regularization imposed during the modelling procedure.

Figure 1. (a) The global spatial distribution of the paleomagnetic sediment records (blue diamonds) and volcanic and archeomagnetic data covering the past 120 ka (with green dots for data older than 10 ka and Holocene data with red dots). (b) Temporal distributions of declination, inclination, and RPI of sediment magnetic records, and archeomagnetic and volcanic data with the three components presented together because of the small number of data. Different width binning is used in order to present the temporal data distribution on one plot; note the y axis label. (c) Sum of data kernels indicating sampling of the core-mantle boundary by all components and all data types; sediment, volcanic, and archeomagnetic data from the semidependent data set are used to build the GGF100k model. RPI = relative paleointensity.

How do the authors explain that the figure differs so strikingly from the figures illustrating the numerical simulations?

The simulations span longer time periods and are much more detailed and higher resolution in both space and time than GGF100k. We have further emphasised this in the text (lines 84-93). We have also modified the figure caption to describe the time periods spanned by the simulations and to note that that time-series in the right-hand column are centred on the peak rate of change.

Some further perspective is supplied by the new Supplementary Figure 8, which shows the equivalent to the lowermost panel of Figure 1, but for LSMOD.2. The higher apparent resolution of LSMOD.2 is evident here, particularly in the geographic distribution of peak angular rates of change. High values occur in both GGF100k and LSMOD.2 in the Western Pacific and Eurasia, but there are distinct differences elsewhere with LSMOD.2 generally exhibiting higher overall values in distinct locations. These reflect the overall shorter time span and smaller number of records in LSMOD.2 combined with

adjustments to the chronology and regional averaging and removal of some of the records prior to modelling. This allowed for a more aggressive strategy in minimizing misfit to the observations in LSMOD.2. Future developments in paleofield modelling with improved data sets are likely to further illuminate which regions actually have the very highest rates of change, but here we have chosen to focus on GGF100k, which provides a conservative analysis for the past 100 kyr.

Document changes: We refer to this supplementary information and figure on line 92-3 of the main text.

If we accept that we have no observational evidence of very rapid changes (on the order of $10^\circ/\text{yr}$) not associated with reversals or excursions, do the authors mean that we should expect that such changes occur, on the basis of their numerical simulations? I have several questions related to this point:

- in the abstract: directional change reach 10 deg. per year even in times of stable polarity: can you illustrate this point ?

The rates of change obtained from GGF and LSMOD.2 were noted on line 214 of the manuscript. We have removed the phrase “even in times of stable polarity” in response to the comment above.

- l.104-106: large values are not necessarily associated with low values of λ_D , can you point examples to the reader ?

In Figure 1 and Supplementary Figure 1 the black lines show the dipole latitude λ_D , which in the case of GGF100k barely deviates from the geographic pole. The VGP latitude shown in purple is another matter. What is obvious is that large VGP deviations do not necessarily coincide with low dipole latitude, but seem to become possible when field strength drops. We have added a sentence to this effect at line 113.

- l 229-230: ‘These results suggest that extremal intensity and directional variations reflect different physical processes at the top of the core.’ Is it correct to conclude also that extremal directional variations are much rarer than extremal intensity events?

While this may be the case we are reluctant to speculate as the required statistical analysis of rapid intensity changes would be a significant undertaking that is outside the scope of the current paper. A direct comparison would also depend on the precise definitions of what constitutes extreme changes in direction and intensity, which cannot be done without robust statistical analysis. This would certainly be an interesting avenue for future work.

Supplementary figure 1: explain how you have selected the 20 thousand years period that is shown and give the total duration of the simulation.

The 20 thousand year focus for the figures is chosen with the most extreme rate of directional change as the mid-point. This information has been added to the caption. Durations have been added to Supplementary Table 1.

REVIEWERS' COMMENTS:

Reviewer #3 (Remarks to the Author):

The authors have answered all my comments appropriately. Their paper offers perspective for future works. I recommend publication in present form.